# Effects of Antibiotic De-Escalation on Outcomes in Severe Community-Acquired Pneumonia: An Inverse Propensity Score-Weighted Analysis

**DOI:** 10.3390/antibiotics14070716

**Published:** 2025-07-17

**Authors:** Diego Viasus, Gabriela Abelenda-Alonso, Juan Bolivar-Areiza, Carlota Gudiol, Jordi Carratalà

**Affiliations:** 1Health Sciences Division, Faculty of Medicine, Hospital Universidad del Norte, Universidad del Norte, Barranquilla 080003, Colombia; 2Infectious Disease Department, Bellvitge University Hospital, Bellvitge Biomedical Research Institute (IDIBELL), L’Hospitalet de Llobregat, 08907 Barcelona, Spain; gabelenda@bellvitgehospital.cat (G.A.-A.); cgudiol@bellvitgehospital.cat (C.G.); jcarratala@bellvitgehospital.cat (J.C.); 3Center for Biomedical Research in Infectious Diseases Network (CIBERINFEC), Instituto de Salud Carlos III, 28029 Madrid, Spain; 4Faculty of Medicine, Universidad de la Sabana, Bogotá 250001, Colombia; jabolivara@gmail.com; 5Clinical Sciences Department, Faculty of Medicine and Health Sciences, University of Barcelona, 08007 Barcelona, Spain

**Keywords:** antibiotic de-escalation, community-acquired pneumonia, duration of intravenous antibiotic therapy, length of hospital stay, propensity score, mortality

## Abstract

**Objective**: This study aimed to assess the effect of antibiotic de-escalation on 30-day mortality, duration of intravenous (IV) antibiotic therapy and length of hospital stay (LOS) in severe community-acquired pneumonia (sCAP). **Methods**: We performed a retrospective analysis of prospectively collected data from a cohort of adults diagnosed with sCAP and microbiologically confirmed etiology between 1995 to 2022. Two distinct time points of the de-escalation were analyzed: 3 and 6 days post-admission, corresponding, respectively, to the availability of microbiological results and the median time to clinical stability. Inverse propensity score-weighted binary logistic regression was used to adjust for potential confounders. **Results**: A total of 398 consecutive cases of sCAP were analyzed. No significant differences were observed between the de-escalation and non-de-escalation groups in terms of age, sex, comorbidities, or severity-related variables (such as impaired consciousness, shock, respiratory failure, or multilobar pneumonia). Patients in the de-escalation group had lower rates of leukopenia, bacteremia and empyema, and less need for mechanical ventilation, with variations depending on the timing of de-escalation. After adjusting for confounding factors in an inverse propensity score-weighted analysis, de-escalation within 3 or 6 days after admission was not associated with increased mortality risk (adjusted odds ratio [aOR] 1.48, 95% confidence interval [CI] 0.29–7.4; *p* = 0.63, and aOR 0.57, 95% CI 0.14–2.31, *p* = 0.43, respectively). Similar findings were observed for prolonged LOS. However, antibiotic de-escalation was related to a lower risk of prolonged IV antibiotic. **Conclusions**: Antibiotic de-escalation in microbiologically confirmed sCAP did not negatively impact clinical outcomes, supporting the safety of this strategy for optimizing antibiotic use in this serious infection.

## 1. Introduction

Patients with severe community-acquired pneumonia (sCAP) require admission to the intensive care unit (ICU) due to the risk of organ dysfunction, which may notably worsen prognosis [1,2]. The diagnostic criteria for sCAP are outlined in the 2007 Infectious Diseases Society of America (IDSA) and American Thoracic Society (ATS) consensus guidelines for the treatment of CAP in adults [3]. Studies have documented a high annual incidence of sCAP, with 145 cases per 100,000 adults in the US and 3.24 cases per 1000 person-years in central Australia. This high incidence has been linked to factors such as advanced age, sex, ethnicity, comorbid conditions, and lower socioeconomic status [4,5]. sCAP carries a markedly elevated mortality risk, with reported rates ranging from 8% to 38% [1,2,4].

Current guidelines recommend broad-spectrum empirical antibiotics for sCAP to cover the most common pathogens [3]. The use of antimicrobial stewardship strategies is strongly encouraged so as to optimize patient outcomes, reduce antibiotic resistance, and minimize toxicity. In this context, antibiotic de-escalation based on microbiological results is advised [3,6]. While de-escalation has been shown to be safe in non-sCAP, its role and safety in severe CAP have been little explored. Hopkins et al. [7] studied patients with negative microbiological results who de-escalated from combination β-lactam and macrolide therapy to β-lactam monotherapy. Of note, studies assessing the effects of de-escalation in critically ill patients with sepsis have generally included only small subsets of patients with respiratory infections, encompassing both community- and hospital-acquired pneumonia [8].

This study aimed to assess the effect of antibiotic de-escalation on clinical outcomes in patients with microbiologically confirmed sCAP. The primary objective was to determine whether de-escalation affects 30-day mortality. Secondary objectives included comparing the duration of intravenous (IV) antibiotic therapy and length of hospital stay (LOS) between patients who underwent de-escalation and those who did not.

## 2. Results

A total of 398 consecutive sCAP cases were analyzed (Figure 1). *Streptococcus pneumoniae* was the most frequently identified pathogen, found in 82.4% of cases (328 patients), followed by *Legionella pneumophila* (39 cases), *Haemophilus influenzae* (24 cases), *Moraxella catharralis* and *Klebsiella pneumoniae* (3 cases each one) and *Pseudomonas aeruginosa* (1 case). Ceftriaxone was the most commonly used empirical antibiotic, administered in approximately 80% of cases. The median duration of IV antibiotic therapy was 7 days (IQR, 4–11 days) and the median LOS was 11 days (IQR, 8–19 days). The median time to clinical stability was 6 days (IQR, 4–12 days). Among the study population, 39 patients (9.7%) underwent antibiotic de-escalation within the first 3 days of hospitalization, while 96 patients (24.1%) were de-escalated by day 6. The proportion of patients de-escalated by day 3 did not change significantly over the study period, except in the earliest interval (1995–2001: 2.9%; 2002–2008: 9.6%; 2009–2015: 13.9%; 2016–2022: 10.3%; chi-square for trend *p* = 0.07). In contrast, the proportion de-escalated within the first 6 days of admission increased over successive intervals (1995–2001: 8.8%; 2002–2008: 19.1%; 2009–2015: 37.4%; 2016–2022: 29.3%; chi-square for trend *p* < 0.001). The all-cause 30-day mortality rate, excluding patients who died within the first 72 h, was 12.3% (49 patients). Mortality did not change significantly over the study period (1995–2001: 14.7%; 2002–2008: 12.7%; 2009–2015: 9.6%; 2016–2022: 13.8%; chi-square for trend *p* = 0.61).

Table 1 shows the characteristics of all patients, along with comparisons between the de-escalation and non-de-escalation groups at both time points (3 and 6 days post-admission). No significant differences were observed between groups regarding age, sex, or comorbidities. Patients in the de-escalation group had lower rates of leukopenia, bacteremia, empyema, mechanical ventilation, and ICU admission, with variations depending on the time point of de-escalation. No significant differences were found between groups in terms of impaired consciousness, tachycardia, tachypnea, shock, respiratory failure, multilobar pneumonia, and empiric antibiotic therapy. At the 3-day time point, the proportion of patients achieving clinical stability was comparable between groups, whereas differences emerged at the 6-day time point.

Table 2 shows the outcomes of the study groups at both time points. All-cause 30-day mortality rates were similar between de-escalated and non-de-escalated patients. The duration of IV antibiotic therapy and the proportion of patients receiving prolonged IV therapy (beyond the median duration) were lower in the de-escalation group. LOS and the proportion of patients with a prolonged LOS (>11 days) were also lower in the de-escalation group. The frequency of adverse drug reactions (allergies, rashes, and phlebitis) was similar in the two groups.

Variables associated with antibiotic de-escalation included in the propensity scores were age (over 65 years), comorbidities, leukopenia, bacteremia, mechanical ventilation, septic shock/hypotension, pneumococcal pneumonia and clinical stability at the time of de-escalation. The *p*-value of the Hosmer–Lemeshow test was 0.97 and the area under the ROC curve was 0.79 (95% CI 0.70–0.88) for the 3-day de-escalation model and 0.41 and 0.78 (95% CI 0.68–0.87), respectively, for the 6-day de-escalation model. In the landmark analysis and using inverse probability weighting based on the propensity score, antibiotic de-escalation was not associated with an increased risk of all-cause 30-day mortality or prolonged LOS. However, antibiotic de-escalation was related to a lower risk of prolonged IV antibiotic (Table 3). To address potential variability in clinical management practices during the early study period, particularly the lower rates of antibiotic de-escalation, a post hoc sensitivity analysis was conducted excluding patients admitted during the first quartile (1995–2001). This analysis yielded comparable results (mortality: aOR 1.90, 95% CI 0.37–9.59, *p* = 0.43 for de-escalation on day 3; and aOR 0.58, 95% CI 0.11–2.94, *p* = 0.51 for de-escalation on day 6).

## 3. Discussion

This study, which focused on patients with microbiologically confirmed sCAP, demonstrates that antibiotic de-escalation did not negatively affect relevant clinical outcomes. Specifically, de-escalation was not associated with increased all-cause 30-day mortality and prolonged LOS.

In our cohort of patients with sCAP, *S. pneumoniae* was the predominant pathogen (82.4%), consistent with other studies that also identify pneumococcus as the main etiology in these patients [9]. We documented a significant difference in the frequency of pneumococcal CAP between the de-escalation and non-de-escalation groups, justifying its inclusion in our multivariate propensity score model. Furthermore, the most common empirical treatment was ceftriaxone (used in approximately 80% of cases), frequently combined with a macrolide or fluoroquinolone. This is consistent with guideline-recommended regimens, which indicate treatment with β-lactam plus macrolides or fluoroquinolones for sCAP [10]. Furthermore, macrolide treatment durations of 3 to 5 days have been recommended in the context of de-escalation therapy and anti-inflammatory properties, but guidelines indicate that further studies are needed to establish the appropriate duration of these therapies [2]. One study showed that the use of macrolides in sCAP reduces complications, even in pneumococcal infections resistant to these antibiotics [11]. Similarly, the addition of clarithromycin for 7 days in the ACCESS trial achieved early clinical anti-inflammatory responses and decreased the need for mechanical ventilation in subgroups of hospitalized patients with CAP [12]. Notably, we found no significant differences in the empirical antibiotic therapies used between the de-escalated and not de-escalated groups. Moreover, the proportion of patients undergoing de-escalation increased during the study period. This likely reflects a growing awareness among physicians in recent decades about the benefits of de-escalation strategies in the absence of formal hospital policy decisions on antimicrobial de-escalation at our institution.

Despite the growing interest in antibiotic de-escalation, evidence supporting its safety and benefit in sCAP remains limited. Previous studies have typically involved small patient cohorts and explored objectives that differ from those of the present study. In this regard, a retrospective cohort study of 94 ICU patients with sCAP found no significant difference in in-hospital mortality when de-escalation was performed from a combination of β-lactam and azithromycin therapy to β-lactam monotherapy after negative multiplex PCR results [7]. Nevertheless, de-escalation was associated with shorter ICU and hospital stays. Additionally, research on de-escalation in non-sCAP populations, including those with negative cultures, pneumococcal pneumonia, or bacteremia, has consistently shown that de-escalation does not increase the risk of mortality [13,14,15].

There are still some gaps in the evidence concerning antibiotic de-escalation that require further investigation. The timing of de-escalation varies across studies, with interventions often implemented after obtaining microbiological cultures or after a predefined period (e.g., 4, 5, or 7 days following hospital admission) [14,15,16]. Furthermore, a standardized definition of antibiotic de-escalation is lacking, with notable variations in its application across studies. Some studies define it as a reduction in the number of antibiotics, while others focus on narrowing the spectrum of coverage (e.g., switching from broad-spectrum to more targeted agents). Its application is further influenced by institutional protocols, the resources available, and local epidemiological patterns. Challenges include the need for quick and accurate microbiological test results, and the risk of treatment failure if carried out too early or with limited information. Standardizing the definition of de-escalation and validating it systematically will further enhance its benefits for both individual patient care and public health [17]. Moreover, various factors have been identified as influencing the decision to de-escalate antibiotics, including clinical stability, chronic kidney disease, ICU admission, pneumococcal pneumonia, Gram-negative pneumonia, and positive microbiological findings [18]. The impact of these factors on the clinical outcomes of patients undergoing antibiotic de-escalation requires further investigation.

In our study, we assessed the effects of de-escalation at two key time points: 3 days and 6 days after hospital admission for sCAP. Day 3 was chosen because microbiological results are generally available at this time, and day 6 because six days was the median time for clinical stability in our cohort. At both time points, we found that de-escalation was not associated with worse outcomes. Notably, de-escalation within the first 3 days of admission is associated with less exposure to broad-spectrum antibiotics, thereby reducing the potential for “collateral damage” caused by antibiotics.

Clinical stability has been identified as a critical factor in the decision to de-escalate antibiotic therapy in clinical practice [18]. Van Heijl et al. [19] demonstrated that clinical stability mediates the effect of de-escalation on mortality, suggesting that patients who achieve stability earlier may benefit more from de-escalation. In our cohort, the median time to clinical stability (6 days, IQR 4–8) was consistent with findings from other studies. Torres et al. [9] reported a similar median time to clinical stability of 5 days (IQR 3–7) in patients with sCAP. Furthermore, in our study, the proportion of patients who achieved clinical stability within the first 3 days was similar in the de-escalation and non-de-escalation groups, suggesting that clinical stability did not significantly influence the decision to de-escalate during the early hospitalization period. As such, clinical stability was included in our propensity score model to account for its potential confounding effects. Other factors related to de-escalation were leukopenia, bacteremia, mechanical ventilation, pneumococcal pneumonia, and empyema. Interestingly, other factors commonly associated with more severe disease, such as respiratory failure, shock, or altered consciousness, did not differ significantly between the two groups.

While our study provides valuable evidence on the safety and efficacy of antibiotic de-escalation in sCAP, several limitations must be considered. The study excluded patients with negative microbiological tests, viral etiology, or co-infection (viral and bacterial pathogens), circumstances that may limit the generalizability of the findings. However, our goal was to provide a focused analysis of what occurs in clinical practice where physicians de-escalate when microbiological information becomes available. Due to the observational design of this study, there is potential for bias, particularly due to the lack of randomization. It is possible that de-escalation was implemented more frequently in patients with less severe sCAP, and that other residual confounding cannot be completely ruled out. However, the use of landmark analysis and inverse probability of treatment weighting helps mitigate these limitations by balancing baseline characteristics across groups. These methods improve causal inference by reducing the influence of confounding variables, allowing for a more reliable estimation of the treatment effect. Additionally, certain factors related to organ dysfunction and prognosis in sCAP, such as the development of acute renal failure or inflammatory markers, were not recorded, potentially affecting the results. Moreover, the non-inclusion of minor criteria for sCAP such as platelet count or blood urea nitrogen levels may have resulted in the exclusion of some patients who would otherwise have been eligible. Finally, the relatively small sample size limits the statistical power of our findings should be considered when interpreting the results. Future studies should include larger populations through multicenter collaboration to allow for subgroup analyses and validate findings in different clinical settings.

## 4. Materials and Methods

This retrospective cohort study analyzed prospectively collected data from consecutive hospitalized patients with CAP from 1995 to 2022. The study was conducted at Bellvitge-IDIBELL University Hospital, a 700-bed tertiary care center in Barcelona, Spain. Institutional review board approval was obtained (PR140/20). Methods and findings are reported in accordance with the Strengthening the Reporting of Observational Studies in Epidemiology (STROBE) guidelines (Appendix A) [20].

### 4.1. Study Population

All patients aged 18 years or older with a clinically, radiographically, and microbiologically confirmed diagnosis of sCAP, as defined by the 2007 IDSA/ATS consensus guidelines [3], were included. sCAP was defined by the presence of either one major criterion (need for invasive mechanical ventilation or septic shock requiring vasopressors) or at least three minor criteria (respiratory rate ≥ 30 breaths per minute, PaO_2_/FiO_2_ ratio < 250, multilobar infiltrates, confusion or disorientation, white blood cell count < 4000/mm^3^, hypothermia and hypotension requiring aggressive fluid resuscitation). Patients who died within 72 h of hospital admission, those already receiving targeted antimicrobial therapy or with pathogens resistant to the antibiotics used in the de-escalation strategy, those without a microbiological etiology or with viral or co-infection pneumonia, and cases with missing data on antibiotic de-escalation were excluded.

For analysis, patients were categorized into two groups: those who underwent antibiotic de-escalation and those who did not. The effects of antibiotic de-escalation were assessed at two time points—3 days and 6 days after admission. The first time point was selected because microbiological test results are typically available by this time, while the second corresponded to the median time to clinical stability in this cohort.

### 4.2. Study Outcomes

The primary outcome was all-cause mortality within 30 days of hospitalization. Secondary outcomes included duration of IV antibiotic therapy and LOS. The duration of IV antibiotic therapy was recorded from the first administered dose until discontinuation, due either to a switch to oral therapy or to completion of the antibiotic regimen. Prolonged IV antibiotic therapy was defined as treatment exceeding the median duration in days. LOS was measured in days from hospital admission to discharge, with prolonged LOS defined as a stay exceeding the cohort’s median value.

### 4.3. Definitions

Empirical antibiotic treatment followed hospital guidelines recommending a β-lactam agent (such as ceftriaxone or amoxicillin/clavulanate), either alone or combined with a macrolide or a fluoroquinolone. Combination therapy was specifically advised for patients with sCAP or when there was clinical suspicion of *Legionella* spp. or other atypical pathogens. De-escalation was defined as the narrowing of the initial β-lactam antimicrobial coverage to penicillin, amoxicillin, or amoxicillin/clavulanate in the case of pneumococcal or *H. influenzae* pneumonia; to a cefepime, ceftazidime or piperacillin/tazobactam in *P. aeruginosa* pneumonia; and to amoxicillin/clavulanate in *K. pneumoniae* and *M. catarrhalis* pneumonia. Continuation of combination therapy of a β-lactam with a macrolide or fluoroquinolone was not considered non-de-escalation, as these agents are recommended for sCAP due to their immunomodulatory effects [2,9,11,12]. Antimicrobial de-escalation was also considered when the initial empirical antimicrobial therapy was narrowed to a fluoroquinolone or macrolide in cases of Legionella pneumonia. During the study period, in the absence of a formal hospital policy, decisions about antimicrobial de-escalation were made at the discretion of the attending physicians. Time to clinical stability was assessed by calculating the number of days from admission until the patient met the criteria defined for stability. These criteria, as characterized in a previous study [21], were evaluated daily throughout the hospital stay.

### 4.4. Microbiological Evaluation

The standard microbiological work-up at admission included sputum culture (requiring a high-quality specimen with a predominant Gram stain morphotype), two sets of blood cultures, and cultures of normally sterile fluids (e.g., pleural fluid). Additionally, urinary antigen detection for *S. pneumoniae* and *L. pneumophila* serogroup 1 was performed using rapid immunochromatographic assays. Microbiological results were used to guide decisions on antibiotic de-escalation. All decisions regarding antibiotic treatment were taken by the attending physicians.

### 4.5. Statistical Analysis

Differences between the de-escalation and non-de-escalation groups were analyzed using the Student *t* test or Mann–Whitney U test for continuous variables, as appropriate, and the chi-square or Fisher exact test for categorical variables. Moreover, given the extended duration of the study, the cohort was divided into quartiles based on the year of hospital admission, and trends over time in key variables (antibiotic de-escalation and mortality) were explored using the chi-square test for trend. To create the propensity score for each patient, the probability that a patient had been de-escalated was assessed with a multivariable analysis which included factors that might influence the decision to de-escalate antibiotic treatment (i.e., variables that were statistically significant in the univariate analysis and variables with clinical relevance). Model fit was assessed with the Hosmer–Lemeshow test, and the discriminatory power was evaluated by the area under the receiver operating characteristic (ROC) curve. In addition, to determine the effect of antibiotic de-escalation on outcomes, a binary logistic inverse propensity score-weighted analysis was performed to adjust for confounders. Since early mortality precludes the opportunity for de-escalation, a landmark analysis was employed to account for immortal time bias when assessing the effect of de-escalation within the first 6 days of hospitalization. The exclusion of patients who died within the first 3 days of hospitalization minimized the risk of immortal time bias in the analysis of early (≤3 days) antibiotic de-escalation [22]. Finally, a post hoc sensitivity analysis was conducted in the binary logistic inverse propensity score-weighted analysis by excluding patients admitted during the first quartile, to assess the robustness of the association between antibiotic de-escalation and outcome, and to account for early period variability in clinical management practices. The SPSS software package was used for statistical analyses (version 20; Chicago, IL, USA). All tests were two-tailed, and a *p*-value < 0.05 was considered statistically significant.

## 5. Conclusions

We found that antibiotic de-escalation in patients with sCAP did not increase the risk of 30-day mortality or a prolonged LOS. Given its potential for reducing the unnecessary use of broad-spectrum antibiotics, de-escalation within 3 days of hospital admission based on microbiologically confirmed etiology should be considered a feasible and effective strategy in the clinical management of sCAP.

## Figures and Tables

**Figure 1 antibiotics-14-00716-f001:**
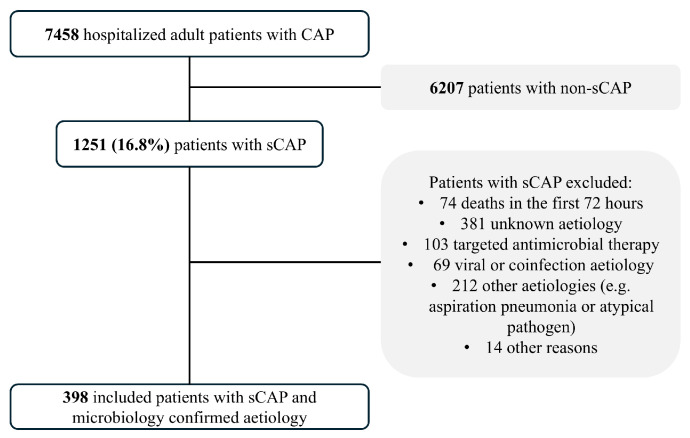
Flowchart of patients with severe community-acquired pneumonia included in the study.

**Table 1 antibiotics-14-00716-t001:** Characteristics of patients with sCAP by study group.

	All Patients	De-Escalation on Day 3		De-Escalation on Day 6	
		De-Escalation Group	Non-De-Escalation Group	De-Escalation Group	Non-De-Escalation Group
	(n = 398)	(n = 39)	(n = 359)	(n = 96)	(n = 302)
** *Demographic data* **					
Age (years), median (IQR)	66 (53–77)	68 (55.5–77)	66 (53–77)	69 (55.5–77.5)	65 (53–77)
Male sex	255 (64.1)	25 (64.1)	230 (64.1)	67 (69.8)	188 (62.3)
Current/former smoker	274 (68.8)	30 (76.9)	244 (68)	73 (76)	201 (66.6)
Pneumococcal vaccine within last 5 years	59 (14.8)	11 (31.4)	48 (15.4)	19 (22.1)	40 (15.4)
** *Comorbid conditions* **					
Chronic pulmonary disease	114 (28.6)	12 (30.8)	102 (28.4)	31 (32.3)	83 (27.5)
Chronic heart disease	92 (23.1)	9 (23.1)	83 (23.19	27 (28.1)	65 (21.5)
Diabetes mellitus	91 (22.9)	10 (25.6)	81 (22.6)	26 (27.1)	65 (21.5)
** *Clinical features* **					
Hypothermia	37 (9.3)	4 (10.3)	33 (9.2)	12 (12.5)	25 (8.3)
Tachycardia (≥100 beats/min)	276 (69.3)	24 (61.5)	252 (72.8)	67 (69.8)	209 (72.3)
Tachypnea (≥24 breaths/min)	288 (72.4)	31 (79.5)	257 (71.6)	72 (75)	216 (71.5)
Impaired consciousness	132 (33.2)	12 (30.8)	120 (33.4)	28 (29.2)	104 (34.4)
Septic shock	47 (11.8)	3 (7.7)	44 (12.3)	11 (11.5)	36 (11.9)
Empyema	24 (6)	0 (0)	24 (6.7)	1 (1)	23 (7.6) *
** *Laboratory and radiographic findings* **					
Leukopenia	54 (13.6)	1 (2.6)	53 (14.8) *	13 (13.5)	41 (13.6)
Respiratory failure (PaO_2_/FiO_2_ < 250)	222 (55.8)	25 (64.1)	197 (54.9)	51 (53.1)	171 (56.6)
Multilobar pneumonia	270 (67.8)	26 (66.7)	244 (68)	59 (61.5)	211 (69.9)
Bacteremia	123 (30.9)	6 (16.2)	117 (34.8) *	24 (26.4)	99 (35.1)
Pneumococcal pneumonia	328 (82.4)	19 (48.7)	309 (86.1) *	64 (66.7)	264 (87.4) *
** *Empiric antibiotic therapy* **					
Ceftriaxone	307 (77.1)	31 (79.5)	276 (76.9)	73 (76)	234 (77.5)
Macrolides or quinolones	250 (62.8)	23 (59)	227 (63.2)	58 (60.4)	192 (63.6)
** *Complications* **					
Mechanical ventilation	103 (25.9)	4 (10.3)	99 (27.7) *	11 (11.6)	92 (30.5) *
ICU admission	170 (42.7)	9 (23.1)	161 (44.8) *	23 (24)	147 (48.7) *
** *Time to clinical stability (days), median (IQR)* **	6 (4–12)	5.5 (3–9)	6 (4–14)	5 (3–8)	7 (4–14.5) *
Clinical stability		8 (25.8)	49 (19.4)	47 (60.3)	74 (36.5) *

sCAP, severe community-acquired pneumonia; IQR, interquartile range; ICU, intensive care unit. All data are presented as number (%) unless otherwise indicated. * *p*-value < 0.05.

**Table 2 antibiotics-14-00716-t002:** Crude outcomes of patients with sCAP stratified by study group.

		De-Escalation on Day 3		De-Escalation on Day 6	
	All Patients	De-Escalation Group	Non-De-Escalation Group	De-Escalation Group	Non-De-Escalation Group
	(n = 398)	(n = 39)	(n = 359)	(n = 96)	(n = 302)
** *Primary outcome* **					
All-cause 30-day mortality	49 (12.3)	3 (7.7)	46 (12.8)	7 (7.3)	42 (13.9)
** *Secondary outcomes* **					
LOS (days), median (IQR)	11 (8–19)	10 (5.5–13.5)	12 (8–20)	8 (5.5–13.5)	13 (8–21) *
LOS above the median	196 (49.2)	17 (43.6)	179 (50.6)	31 (32.3)	165 (55.6) *
IV antibiotic therapy (days), median (IQR)	7 (4–11)	3 (2–5)	7 (5–13) *	4 (3–6)	8 (5–13) *
IV antibiotic therapy above the median	175 (44)	6 (15.8)	169 (49) *	17 (18.3)	158 (54.5) *
Adverse drug reactions **	32 (8)	1 (2.6)	31 (8.6)	4 (4.2)	28 (9.3)

sCAP, severe community-acquired pneumonia; IQR, interquartile range; LOS, length of hospital stay; IV, intravenous. All data are presented as number (%) unless otherwise indicated. * *p* value < 0.05. ** Adverse drug reactions include allergies, rashes, and phlebitis.

**Table 3 antibiotics-14-00716-t003:** Impact on outcomes of antibiotic de-escalation in patients with sCAP: a binary logistic inverse propensity score-weighted analysis.

	De-Escalation on Day 3			De-Escalation on Day 6 *		
	aOR	CI 95%	*p*-Value	aOR	CI 95%	*p*-Value
** *Primary outcome* **						
All-cause 30-day mortality	1.48	(0.29–7.40)	0.63	0.57	(0.14–2.31)	0.43
** *Secondary outcomes* **						
LOS above the median	0.75	(0.38–1.47)	0.41	0.65	(0.32–1.33)	0.24
IV antibiotic therapy above the median **	0.22	(0.06–0.74)	0.01	0.39	(0.17–0.85)	0.01

sCAP, severe community-acquired pneumonia; aOR, adjusted odds ratio; CI, confidence interval; LOS, length of hospital stay; IV, intravenous. * To mitigate immortal time bias, a landmark analysis was performed on de-escalation within the first 6 days of hospitalization. The analysis excluded 9 patients who died during this initial period. ** Similar findings were observed after excluding patients with empyema (aOR 0.25, CI 95% 0.07–0.76, *p*-value 0.02 for de-escalation on day 3; and aOR 0.13, CI 95% 0.06–0.27, *p*-value < 0.001 for de-escalation on day 6).

## Data Availability

The raw data supporting the conclusions of this article will be made available by the authors on request.

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
