# Peer review of "Effects of Antibiotic De-Escalation on Outcomes in Severe Community-Acquired Pneumonia: An Inverse Propensity Score-Weighted Analysis"

_antibiotics, 2025, doi:10.3390/antibiotics14070716_

Round 1

Reviewer 1 Report

Comments and Suggestions for Authors

This manuscript investigates antibiotic de-escalation in severe community-acquired pneumonia (sCAP), examining 30-day mortality, IV antibiotic duration, and hospital stay. The study employs robust methodology with inverse probability weighting to control confounding, addressing a clinically relevant question with clear, practical conclusions. However, several limitations require attention for methodological strengthening.

Limitations

The study faces several methodological constraints that limit its impact. The small de-escalation cohort (39 patients at day-3) compromises statistical power and conclusion robustness. Retrospective design introduces potential selection bias despite statistical adjustments, with residual confounding remaining a concern. Generalizability is restricted by excluding patients with negative cultures, viral infections, or mixed infections, limiting applicability to broader clinical populations. Additionally, the study provides limited coverage of diverse antibiotic combinations and clinical scenarios commonly encountered in practice, reducing real-world applicability.

Improvement Recommendation

Future studies should prioritize expanding the patient population, particularly de-escalation cases, through multi-center collaboration to enhance statistical power. Comprehensive sensitivity analyses and pathogen-specific or severity-based subgroup analyses would validate findings across different clinical contexts. The research would benefit from detailed guidance on clinical indicators (biomarkers, response parameters) that inform de-escalation decisions, providing standardized decision criteria for practitioners. Implementation focus should include explicit clinical guidelines, practical barriers, and decision-making frameworks to facilitate real-world application. Finally, acknowledging unmeasured clinical variables (kidney function, inflammatory markers) that may influence outcomes would improve methodological transparency and guide future research priorities.

Author Response

Reviewer 1

This manuscript investigates antibiotic de-escalation in severe community-acquired pneumonia (sCAP), examining 30-day mortality, IV antibiotic duration, and hospital stay. The study employs robust methodology with inverse probability weighting to control confounding, addressing a clinically relevant question with clear, practical conclusions. However, several limitations require attention for methodological strengthening.

REPLY: We sincerely appreciate the reviewer’s evaluation of our manuscript. We are grateful for the recognition of our methodological approach, including the use of inverse probability weighting to address confounding, and for acknowledging the clinical relevance and clarity of our conclusions.

Comments 1.1:  Limitations

The study faces several methodological constraints that limit its impact. The small de-escalation cohort (39 patients at day-3) compromises statistical power and conclusion robustness. Retrospective design introduces potential selection bias despite statistical adjustments, with residual confounding remaining a concern. Generalizability is restricted by excluding patients with negative cultures, viral infections, or mixed infections, limiting applicability to broader clinical populations. Additionally, the study provides limited coverage of diverse antibiotic combinations and clinical scenarios commonly encountered in practice, reducing real-world applicability.

REPLY 1.1: We appreciate the reviewer’s careful assessment of the methodological limitations of our study, and we fully acknowledge these important points, which are also discussed in the manuscript (lines 207–227). We recognize that the relatively small number of patients in the de-escalation group limits the statistical power and the robustness of our conclusions. This limitation is inherent to the retrospective nature of our study and the real-world incidence of de-escalation in this clinical context. To mitigate potential biases, we employed rigorous statistical techniques, including inverse probability of treatment weighting (IPTW) and landmark analysis. Regarding generalizability, we acknowledge that our exclusion criteria—specifically, the exclusion of patients with negative cultures, viral infections, or mixed infections—may restrict the applicability of our findings to a broader clinical population. However, our intention was to focus on a well-defined cohort in which de-escalation decisions are typically made based on microbiological data, reflecting a common scenario in clinical practice (lines 210–212).

Comment 1.2: Improvement Recommendation

Future studies should prioritize expanding the patient population, particularly de-escalation cases, through multi-center collaboration to enhance statistical power. Comprehensive sensitivity analyses and pathogen-specific or severity-based subgroup analyses would validate findings across different clinical contexts. The research would benefit from detailed guidance on clinical indicators (biomarkers, response parameters) that inform de-escalation decisions, providing standardized decision criteria for practitioners. Implementation focus should include explicit clinical guidelines, practical barriers, and decision-making frameworks to facilitate real-world application. Finally, acknowledging unmeasured clinical variables (kidney function, inflammatory markers) that may influence outcomes would improve methodological transparency and guide future research priorities.

REPLY 1.2:  We are grateful to the reviewer for these insightful recommendations to strengthen both the current manuscript and future research in this area. In response, we have revised the Discussion section to explicitly underscore the need for larger, multi-center studies—particularly to increase the number of de-escalation cases and thereby enhance statistical power and generalizability. We agree that collaborative, multi-institutional efforts are essential to capture a broader spectrum of patient populations and clinical practices. Additionally, we now highlight the importance of comprehensive sensitivity analyses, as well as subgroup analyses based on pathogen type and disease severity, to validate findings across diverse clinical contexts (lines 226–227). We have also expanded our discussion of the clinical indicators that inform de-escalation decisions, and we noted some practical challenges that may affect real-world application (lines 175–178). Finally, we acknowledge that unmeasured clinical variables—such as kidney function and inflammatory markers—may influence outcomes. We have noticed the importance of accounting for these factors in future studies to improve methodological transparency and further refine risk stratification (lines 219–222).

Reviewer 2 Report

Comments and Suggestions for Authors

The manuscript describes a retrospective study conducted at a single tertiary care centre in Spain. The study aims to see the impact of antibiotic de-escalation in severe CAP with 30-day mortality as the primary end point. The topic is of interest as it focusses on an important strategy to optimise antibiotic use as a component of antibiotic stewardship. The manuscript well describes the rationale and details on study conduct. Few areas which need to be addressed are:

  1. The study data covered a long period i.e. 1995 – 2022; during this long span of nearly three decades, one assumes reasonable changes in antibiotic prescribing practices depending on emerging resistance and other factors. These should be mentioned in the text.
  2. Methods: a brief account of the various antibiotic de-escalation strategies have been given. But further details on various strategies adopted across different clinical scenarios should be added to give a better picture of the practices. Also, what were the evolving trends in de-escalation strategies in the study setting across the study period.
  3. Were there any hospital-based guidelines for antibiotic de-escalation in the sCAP?
  4. Was there any impact of the different de-escalation strategies on the outcome? In my opinion, adding this data will add value to the study in terms of any such variation if observed, and thus suggesting specific strategies with greater impact over others.
  5. What was the case definition of severe CAP?
  6. The study outcomes have been assessed for de-escalation versus no de-escalation. Other factors such as antibiotic susceptibility patterns, causative organism, specific antibiotic agents etc. also tend to affect the outcome which have not been taken into account during analysis and interpretation. I would suggest the authors to add more data regarding this in the manuscript.
  7. The authors are advised to add the points mentioned above in the manuscript and interpret the additional observations under discussion and conclusion.
Comments on the Quality of English Language

Moderate editing in English language required.

Author Response

Reviewer 2

The manuscript describes a retrospective study conducted at a single tertiary care centre in Spain. The study aims to see the impact of antibiotic de-escalation in severe CAP with 30-day mortality as the primary end point. The topic is of interest as it focusses on an important strategy to optimise antibiotic use as a component of antibiotic stewardship. The manuscript well describes the rationale and details on study conduct.

REPLY: We appreciate the reviewer’s thoughtful evaluation of our manuscript and their recognition of the study’s relevance to antibiotic stewardship in severe CAP. We are grateful for the constructive feedback and recommendations, which have helped us to improve the clarity and quality of the manuscript.

Few areas which need to be addressed are:

Comment 2.1: The study data covered a long period i.e. 1995 – 2022; during this long span of nearly three decades, one assumes reasonable changes in antibiotic prescribing practices depending on emerging resistance and other factors. These should be mentioned in the text.

REPLY 2.1: We thank the reviewer for highlighting the importance of temporal changes in antibiotic prescribing practices over the nearly three-decade study period. In response, we have added detailed information in both the Methods and Results sections to address this issue. Specifically, we divided the cohort into quartiles based on the year of hospital admission and analyzed temporal trends in key variables, including antibiotic de-escalation and mortality, using the chi-square test for trend (lines 74–75, 80–88, 96, and 291–294). Additionally, we have expanded Table 1 to provide more comprehensive information on the antibiotics prescribed during different periods of the study.

 Comment 2.2: Methods: a brief account of the various antibiotic de-escalation strategies have been given. But further details on various strategies adopted across different clinical scenarios should be added to give a better picture of the practices. Also, what were the evolving trends in de-escalation strategies in the study setting across the study period.

REPLY 2.2: We have added information about this topic in the methods section (lines 74-75, 80-88, 96 and 291-294).

Comment 2.3: Were there any hospital-based guidelines for antibiotic de-escalation in the sCAP?

REPLY 2.3: We have added information regarding hospital-based guidelines for antibiotic de-escalation in sCAP to the Methods section (lines 261–265 and 274–276).

Comment 2.4: Was there any impact of the different de-escalation strategies on the outcome? In my opinion, adding this data will add value to the study in terms of any such variation if observed, and thus suggesting specific strategies with greater impact over others.

REPLY 2.4: Thank you for this valuable suggestion. We have analyzed the impact of the two de-escalation strategies (at 3 days and at 6 days) on clinical outcomes and found that both approaches yielded similar results. As noted in the manuscript, de-escalation at 3 days may offer potential advantages, such as reduced antibiotic exposure and less "collateral damage." However, we acknowledge that our sample size limits definitive conclusions, and further research with larger, population-based studies or randomized trials is needed to determine if specific de-escalation strategies confer greater benefit (lines 189–191).

Comment 2.5: What was the case definition of severe CAP?

REPLY 2.5: We have added the case definition of severe CAP (lines 238-242).

Comment 2.6: The study outcomes have been assessed for de-escalation versus no de-escalation. Other factors such as antibiotic susceptibility patterns, causative organism, specific antibiotic agents etc. also tend to affect the outcome which have not been taken into account during analysis and interpretation. I would suggest the authors to add more data regarding this in the manuscript.

REPLY 2.6: We have included information on causative organisms, and specific antibiotic agents (lines 80–88, 96, 244-245, and Table 1) and expanded the discussion accordingly (lines 137–157).

Comment 2.7: The authors are advised to add the points mentioned above in the manuscript and interpret the additional observations under discussion and conclusion.

REPLY 2.7: We have addressed these points in the revised manuscript as detailed in our previous responses.

Reviewer 3 Report

Comments and Suggestions for Authors

 a retrospective analysis of prospectively collected data from a cohort of adults diagnosed with sCAP and microbiologically confirmed etiology.

Abstract [also line 96-99] “However, antibiotic de-escalation was related to a lower risk of prolonged IV antibiotic.” Is self evident  - please remove.

The abstract should state that the cohort was recruited between 1995 to 2022. Please porivde the distribution of year of patient presentation. Moreover, why isn’t year of presentation in the model?

Should empyema be an exclusion as this would require prolonged therapy?

The definition of De-escalation [4.3] is strange and contradicts what you state at line 55/57. E.g. “Continuation of combination therapy of a β-lactam with a macrolide or fluoroquinolone was not considered non-de-escalation,…” many would disagree with this inclusion in the definition. At the very least this requires either a sensitivity test whether this is included in how de-escalation is defined or removal of these patients from the analysis.

Author Response

Reviewer 3

A retrospective analysis of prospectively collected data from a cohort of adults diagnosed with sCAP and microbiologically confirmed etiology.

REPLY:  We sincerely thank the reviewer for their valuable and appreciated comments.

Comment 3.1: Abstract [also line 96-99] “However, antibiotic de-escalation was related to a lower risk of prolonged IV antibiotic.” Is self evident  - please remove.

REPLY 3.1: We have included this information because evaluating the effect of antibiotic de-escalation on the duration of intravenous antibiotic therapy was one of our study objectives.

Comment 3.2: The abstract should state that the cohort was recruited between 1995 to 2022. Please porivde the distribution of year of patient presentation. Moreover, why isn’t year of presentation in the model?

REPLY 3.2: We have added the recruitment period to the abstract (line 22) as requested. We have also included information on the distribution of variables by quartile according to the year of admission (lines 80-88) and a sensitivity analysis excluding patients admitted during the first quartile (lines 122-127 and 291-294), considering the low rate of de-escalation during this period and the fact that the de-escalation rate and mortality were similar during the other quartiles.

Comment 3.3: Should empyema be an exclusion as this would require prolonged therapy?

REPLY 3.3: We excluded patients with empyema from the analysis of prolonged antibiotic therapy (Table 3).

Comment 3.4: The definition of De-escalation [4.3] is strange and contradicts what you state at line 55/57. E.g. “Continuation of combination therapy of a β-lactam with a macrolide or fluoroquinolone was not considered non-de-escalation,…” many would disagree with this inclusion in the definition. At the very least this requires either a sensitivity test whether this is included in how de-escalation is defined or removal of these patients from the analysis.

REPLY 3.4: In hospitalized adults with severe CAP, current IDSA guidelines recommend a β-lactam plus a macrolide or a β-lactam plus a respiratory fluoroquinolone. Both macrolides and fluoroquinolones have demonstrated anti-inflammatory effects. Furthermore, macrolide treatment durations of 3 to 5 days have been recommended in the context of de-escalation therapy and their anti-inflammatory properties, although guidelines indicate that further studies are needed to establish the optimal duration of these therapies. This recommendation is followed in clinical practice at many institutions. Similarly, the addition of clarithromycin in the ACCESS trial resulted in early clinical anti-inflammatory responses and reduced the need for mechanical ventilation in certain subgroups of hospitalized patients with CAP; the duration of macrolide therapy in the ACCESS study was 7 days. Likewise, Restrepo et al. reported an association between macrolide use and reduced 30- and 90-day mortality in severe sepsis due to pneumonia, even among patients with macrolide-resistant pathogens. We have added information about this topic in the Discussion section (lines 132–154).

Round 2

Reviewer 2 Report

Comments and Suggestions for Authors

No further comments.

Reviewer 3 Report

Comments and Suggestions for Authors

Thankyou for attending to the comments